# Optimal Siting and Sizing of Distributed Generation Based on Improved Nondominated Sorting Genetic Algorithm II

**Wei Liu [1],\*, Fengming Luo [2]****, Yuanhong Liu [1,3],\* and Wei Ding [1]**

[1] School of Electrical Information and Engineering, Northeast Petroleum University, Daqing 163318, China; vivi_ting@stu.nepu.edu.cn
[2] Lorentech (Beijing) Co., Ltd., Beijing 100000, China; LFM1220@126.com
[3] Faculty of Engineering and Environment, Northumbria University, Newcastle NE1 8ST, UK
\* Correspondence: liuwei@nepu.edu.cn (W.L.); liu.yuanhong@northumbria.ac.uk (Y.L.)

**Abstract:** With the development of distributed generation technology, the problem of distributed generation (DG) planning become one of the important subjects. This paper proposes an Improved non-dominated sorting genetic algorithm-II (INSGA-II) for solving the optimal siting and sizing of DG units. Firstly, the multi-objective optimization model is established by considering the energy-saving benefit, line loss, and voltage deviation values. In addition, relay protection constraints are introduced on the basis of node voltage, branch current, and capacity constraints. Secondly, the violation constrained index and improved mutation operator are proposed to increase the population diversity of non-dominated sorting genetic algorithm-II (NSGA-II), and the uniformity of the solution set of the potential crowding distance improvement algorithm is introduced. In order to verify the performance of the proposed INSGA-II algorithm, NSGA-II and multiple objective particle swarm optimization algorithms are used to perform various examples in IEEE 33-, 69-, and 118-bus systems. The convergence metric and spacing metric are used as the performance evaluation criteria. Finally, static and dynamic distribution network planning with the integrated DG are performed separately. The results of the various experiments show the proposed algorithm is effective for the siting and sizing of DG units in a distribution network. Most importantly, it also can achieve desirable economic efficiency and safer voltage level.

**Keywords:** distributed generation (DG); INSGA-II; multi-objective optimization; potential crowding distance; static and dynamic planning

## 1. Introduction

In recent years, distributed generation systems using renewable energy technologies such as wind power generation and photovoltaic power generation have become one of the hotspots of research at home and abroad. After the distributed generation (DG) units are connected to the distribution network (DN), the structure, operation mode, and control strategy of the DN will undergo tremendous changes [1,2]. The research shows that DGs provide more flexibility and expansibility for distribution network. When DGs are connected to the distribution network (DN), the performances of DGs are most important for the power quality, reliability, and security of DN. In addition, DGs with renewable energy technology can effectively reduce system line loss and transmission congestion [3–5]. Despite the above advantages, if the placement and sizing of the DGs are improperly selected, it may cause a series of power system safety hazards such as power flow reverse, insufficient voltage stability, and malfunction of the protection device [6,7]. Therefore, the placement and sizing of the DGs has become one of the important subject.

The optimal planning of distributed generation sizing and siting is critical to ensure the operational performance of a distribution network in terms of power quality, voltage stability, reliability and profitability. DG planning problems are often defined as multi-objective and multi-constrained optimization problems. A number of review papers [8–10] have surveyed the optimization techniques for optimal DG planning in power distribution networks. The aforementioned review papers mainly focused on the discussion of various computational methods and metaheuristic algorithms.

Nowadays, industrial systems, such as papermaking, steelmaking, petrochemical, and power generation, are becoming more and more complex. In such cases, data-driven models based on novel nonlinear signal processing and data analysis techniques may provide an attractive alternative [11]. Therefore, it is paramount but challenging to develop effective techniques in modelling, monitoring, and control for complex industrial systems [12,13].

Among many optimization methods, multi-objective heuristic algorithms are the main means to solve multi-objective optimization problems today because they can effectively balance multiple objectives for optimal search [14]. At present, the main objectives of DG planning include the lowest investment cost of DG, the lowest environmental pollution, the optimal voltage quality of the power grid, and the minimum power loss. Partha Kayal et al. aimed to minimize the reactive power loss of the DN, to maximize the system voltage stability, and to establish a multi-objective optimization model for power systems [15]. Although it can effectively reduce the network loss and improve the voltage distribution, planning with only two objectives does not reflect the real condition. R. Li et al. established a multi-objective optimization model considering economy, voltage quality, and network loss and used the distance entropy multi-objective particle swarm optimization algorithm to solve the optimization problem [16]. However, there is a lack of environmental energy-saving factors in economic costs; M. Esmaili and T. Wang et al. also introduced voltage safety margins [17] and pollutant emissions [18] as optimization models, respectively. However, the objective function of optimization varies greatly in the timescale. For example, the voltage and reactive power change rapidly, and the pollutant emission often has a long timescale, which makes it difficult to use a unified model to optimize. Therefore, this paper establishes a multi-objective optimization model for economy, environmental protection, safety, and reliability.

Constraint processing is a key part of engineering optimization problems, and the treatment of constraints is the key to solving constrained multi-objective optimization problems. Commonly used constraint handling methods include rejection of infeasible solutions, penalty functions, and various correction algorithms. In fact, in the iterative process, infeasible solution is always difficult to be rejected, especially in the case of small feasible region, and repeated trial and error will affect the speed of solution; other modified algorithms are always designed for specific problems. The penalty function method [19] is the most classical method to deal with constraints, but the penalty factor has the same weight problem and is not easy to grasp, resulting in large errors in planning results.

In this paper, a new operator is proposed to address the degree of default. This method does not need to set parameters in advance but deals with feasible and infeasible solutions by considering the number and degree of violation of constraints. This method can effectively combine with multi-objective optimization algorithm to improve the diversity of populations and to avoid the algorithm falling into local optima.

At present, the planning methods of multi-objective optimization problems can be divided into weighted single-objective optimization algorithm and multi-objective optimization algorithm. In order to obtain the unique solution, scholars transform the multi-objective into single-objective optimization model by the weight method and solve it by combining with the single-objective optimization algorithm. This method makes the calculation convenient by reducing the dimension of the problem. For example, H. Su introduced improved simulated annealing particle swarm optimization [20] and established a multi-objective model to maximize DG utilization and to minimize system losses and environmental pollution. Finally, the DG injection model is optimized by this algorithm. However, the selection of weights depends on experience. Multi-objective functions are both interrelated and independent. The

subjectivity of weight selection makes the calculation result unsatisfactory. Aiming at the above problems, multi-objective optimization algorithm is introduced to solve the DG planning problem [21]. J. Neale solved the reconfiguration problem of the DN with the integrated DG by using non-dominated sorting genetic algorithm-II (NSGA-II) [22]. This method can effectively improve the operation performance of DN. A. Ameli used the multi-objective particle swarm optimization (MOPSO) [23] with adaptive grid method to optimize the distribution of energy supply system electric vehicle (EV) in DG technology. The results show the effectiveness of MOPSO programming; R. S. Maciel proposed an evolutionary particle swarm optimization algorithm (MEPSO) [24] as a multi-objective optimization algorithm for DG, of which the planning scheme can ultimately improve the objectives. However, in the multi-objective optimization algorithm, there is a common problem; that is, the evaluation information of solution set is insufficient due to crowding distance, which leads to the elimination of potential high-quality solutions in the truncation process. The distribution of the final algorithm planning results is uneven.

In addition, the mutation operator of NSGA-II will gradually lose the ability of local search as the dimension of solution variable increases, so the population updating strategy in the fireworks algorithm [25] is introduced to simultaneously improve the mutation operator and the diversity of solution set. Based on the effective Pareto-optimal set of multi-objective optimization problems, decision makers cannot get a unique solution. Therefore, this paper introduces an unbiased compromise strategy [26] to choose a best compromise from the Pareto-optimal set.

According to the above analysis, a lack of the operation state research planning of DN with the integrated DG considers the relay protection. Actually, when the DG is connected to the DN, the DN that is originally powered by a single system power becomes a network with multiple power structures, causing a change in the magnitude and direction of the short-circuit current during the failure. The action affects the safe and stable operation of the DN. Therefore, this paper increases the short-circuit current constraint condition and compares the result with the optimization result without considering the short-circuit constraint, so as to prove the necessity of considering the relay protection condition. At present, with the development of technology, distributed energy resources are bound to penetrate into the fields of industry, commerce, and urban and rural residents. The problem of DG planning with timing characteristics needs to be solved urgently. Therefore, the paper studies the planning of various examples and plans the static state and dynamic running state of the IEEE 33-bus system separately to provide a reasonable DG planning for decision makers.

The main contributions of this work are as follows:

(1) An improved non-dominated sorting genetic algorithm (INSGA-II) for placement and sizing of DGs is proposed. The violation constrained index and improved mutation operator are proposed to increase the population diversity of NSGA-II, and the uniformity of the solution set of the potential crowding distance improvement algorithm under the 3D objective function is introduced.

(2) The energy-saving benefit of DGs is considered, and a multi-objective optimization model is established.

(3) The restraint condition of relay protection is added; that is, the size of short-circuit current is limited, so that the protection device will not malfunction.

(4) Convergence metric and spacing metric are employed to evaluate the performance of INSGA-II, NSGA-II, and MOPSO.

(5) The effectiveness of the algorithm in different cases is verified.

The proposed algorithm is applied to solve the static and dynamic planning of the DN with the integrated DG units. The rest of this article is organized as follows. Section 2 presents the mathematical programming model. Section 3 briefly introduces the algorithms needed in the planning scheme and proposes INSGA-II. Section 4 provides the comparison of the simulation results of the proposed algorithm in various cases and the performance of the algorithm. Section 5 concludes our work.

## 2. Problem Formulation

### 2.1. Objective Functions

Three objectives should be taken into account when establishing multi-criteria optimization model of DN with the integrated DG units: (1) Improving the energy-saving benefit of the system; (2) reducing system line losses; and (3) reducing the node voltage deviation.

1.  Maximizing annual energy-saving benefit

Maximizing annual energy-saving benefit of DN with the integrated DG units can be expressed as follows:

$$\max f_1 = Z_{\text{cost}}^{\text{NODG}} - Z_{\text{cost}}^{\text{DG}} \tag{1}$$

where $Z_{\text{cost}}^{\text{NODG}}$ is the total annual cost of DN without DGs and $Z_{\text{cost}}^{\text{NODG}}$ is the total annual cost of DN with DGs.

Total annual costs of DN excluding DGs can be expressed as follows:

$$Z_{\text{cost}}^{\text{NODG}} = C_{\text{loss}} + C_{\text{b}} \tag{2}$$

where $C_{\text{loss}}$ is the annual loss of DN without DGs and $C_{\text{b}}$ is the annual purchase cost of DN without DGs.

$$C_{\text{loss}} = \sum_{j=1}^{k} \left( C_{\text{p}} \cdot \tau_{\text{max}} \cdot \Delta P_j \right) \tag{3}$$

where $k$ is the number of branches in the DN; $C_{\text{p}}$ is the unit price of electricity consumed per unit (\$/kWh); $\tau_{\text{max}}$ is the annual maximum load loss hour (h) of the DN; and $\Delta P_j$ is the active power loss (kW) of the $j$th branch.

$$C_{\text{b}} = C_{\text{p}} \cdot P_{\text{load}} \cdot T_{\text{max}} \tag{4}$$

where $P_{\text{load}}$ is the DN total active load and $T_{\text{max}}$ is the DN annual maximum load utilization hours.

The total annual cost of DN with DGs can be expressed as follows:

$$Z_{\text{cost}}^{\text{DG}} = C_{\text{dgm}} + C_{\text{ploss}} + C_{\text{B}} - C_{\text{sub}} \tag{5}$$

where $C_{\text{dgm}}$ is the annual investment and maintenance cost of DGs; $C_{\text{ploss}}$ is the annual cost of DN with DGs (calculated in the same way as $C_{\text{loss}}$); $C_{\text{B}}$ is the annual purchase cost of DN with DGs; and $C_{\text{sub}}$ is the financial subsidy of new energy generation.

$$C_{\text{dgm}} = \sum_{i=1}^{n} \left( \frac{r(1+r)^t}{(1+r)^t - 1} \cdot C_{dgi} + C_{mi} \right) \cdot P_{dgi} \tag{6}$$

where $r$ is the annual interest rate; $t$ is the planning period; $C_{dgi}$ is the $i$th distributed generation equipment investment cost (\$/kWh); $C_{mi}$ is the annual operation and maintenance cost (\$/kWh) for the $i$th distributed generation; and $P_{dgi}$ is the installed capacity (kW) of the $i$th distributed generation.

$$C_{\text{B}} = C_{\text{P}} \cdot \left( P_{\text{load}} - \sum_{i=1}^{n} \lambda_i \cdot S_{dgi} \right) \cdot T_{\text{max}} \tag{7}$$

In order to reflect the benefits of DGs in environmental protection, the government has put forward policy support to DGs, that is, financial subsidies for distributed generation. It can be expressed as follows:

$$C_{\text{sub}} = C_{\text{sp}} \cdot \sum_{i=1}^{n} P_{dgi} \cdot T_{\text{max}} \tag{8}$$

where $C_{sp}$ is the subsidy amount ($) of distributed generation unit.

2. Minimize line losses

The existence of line losses will lead to line heating, which will accelerate the aging of insulated lines, will reduce the insulation of lines, and will ultimately lead to the risk of leakage. Reducing line losses can improve energy efficiency of electrical equipment or processes.

Lowering the line losses can improve the energy utilization efficiency of the energy-using equipment or process, and it is also one of the important measures of energy saving. The objective function can be expressed as follows [27]:

$$\min f_2 = \sum_{i,j \in n} g_{ij}\left(U_i^2 + U_j^2 - 2U_iU_j \cos \theta_{ij}\right) \tag{9}$$

where $n$ is the total number of DN nodes; $g_{ij}$ is the admittance of the branch $(i, j)$; $U_i$ and $U_j$ are the voltage amplitudes of branches $i$ and $j$ respectively; and $\theta_{ij}$ is the voltage phase angle.

3. Minimize the total voltage deviation

With the increase of load, the system voltage stability will deteriorate gradually, even voltage collapse, resulting in a system in danger. Therefore, the voltage deviation is one of the important indexes to evaluate the operation safety and power quality of the system. In addition, the increase of network node voltage can effectively reduce the reactive power loss of the system. The objective function can be expressed as follows:

$$\min f_3 = \sum_{i=0}^{N} \frac{\left|U_i - U_i^{specified}\right|}{U_i^{\max} - U_i^{\min}} \tag{10}$$

where $U_i$ is the node $i$th voltage real value of the DN when DG is incorporated and $U_i^{specified}$ is the voltage rated value. This paper assumes that the rated voltage is 1 (p.u).

### 2.2. Influence of DG on Relay Protection of DN

DN is generally equipped with three-stage current protection. After the DG is connected to the DN, the system changes from single power supply to multiple power supply. When the transmission line is short-circuited, the magnitude and direction of the short-circuit current will change.

Figure 1 shows a typical simple DN. $L_1$ and $L_2$ represent feeders; $T_1$ represents a transformer; $PD_1$, $PD_2$, $PD_3$, and $PD_4$ are protective devices of the corresponding feeders; and DG is connected to the DN from node C.

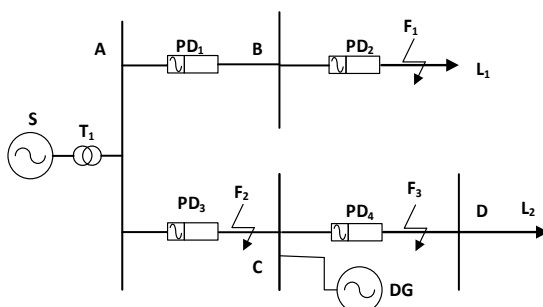

**Figure 1.** Distribution network system diagram of distributed generation (DG).

Assuming that the DG is connected, the short-circuit fault can be divided into the following three cases:

1. The adjacent feeder connected to the DG is short-circuited

When the short-circuit fault $F_1$ occurs at the end of the adjacent feeder, the power source *S* and DG provide the overlapping short-circuit current to $PD_1$ and $PD_2$. Compared without DG connection, the short-circuit current increases and its value may be larger than the original I segment setting value of $PD_1$, which makes $PD_1$ malfunction, resulting in the loss of coordination between $PD_1$ and $PD_2$. Simultaneously, $PD_3$ will also flow through the reverse current provided by DG. When the capacity of DG is larger, the value of reverse current may be larger than the original I segment setting value of $PD_3$, resulting in $PD_3$ malfunction.

2.  The upstream of DG access is short-circuited

When the DG upstream feeder AC is short-circuit fault $F_2$, $PD_4$ can operate normally. However, when $PD_3$ is activated, the island downstream of the DG will form an island operation. At the same time, the DG will always provide short-circuit current to the fault point, which will affect the reliability of the system.

3.  The downstream of DG access is short-circuited

When DG downstream feeder CD short-circuit fault $F_3$ occurs, protections $PD_1$ and $PD_2$ can operate normally. Although the short-circuit current through $PD_3$ is only supplied by the system power *S*, the fault current will be smaller than before due to DG connection, which may cause the backup protection of $PD_3$ to refuse to operate. The short-circuit current through $PD_4$ is provided by the system power *S* and DG together. The increase of short-circuit current may increase the protection distance of $PD_4$, thus losing the selectivity.

When there are multiple DGs in the system, the analysis method is similar. From the above analysis, it can be seen that the access of DG will have adverse effects on the reliability of the relay protection and system of DN, so it is necessary to consider the restraint of relay protection when optimizing the configuration of DG. In the traditional three-stage current protection, the I-stage instantaneous current quick-break protection is the most important, so this paper focuses on the analysis of the impact of DG on the I-stage of the original current protection of the PD in the DN.

*2.3. Constraint Condition*

DG connected to a distribution network has great influence on power flow, voltage distribution, branch current, and power quality of the system. In order to design a reasonable programming model, the following equality constraints and inequality constraints are included in this paper.

1.  Equality constraint

Below are the power flow equations constraints.

$$
\begin{cases}
P_{DGi} - P_{Li} - U_i \sum_{k \in i}^{N_1} U_k (G_{ik} \cos \theta_{ik} + B_{ik} \sin \theta_{ik}) = 0 \\
Q_{DGi} - Q_{Li} - U_i \sum_{k \in i}^{N_1} U_k (G_{ik} \sin \theta_{ik} + B_{ik} \cos \theta_{ik}) = 0
\end{cases}
\tag{11}
$$

where $P_{DGi}$ and $Q_{DGi}$ are the *i*th node active and reactive power of the DGs injected; $P_{Li}$ and $Q_{Li}$ are the *i*th node active and reactive powers of load output; $U_k$ is the voltage amplitude of all *k*th nodes connected to *i*th nodes; $G_{ik}$ is branch conductance; $B_{ik}$ is the branch susceptance; $\theta_{ik}$ is the difference between the voltage angles of *i*th node and *k*th node; and $N_1$ is the number of branches associated with the *i*th node.

2.  Inequality constraints

In order to make the planning scheme meet the power system operation standards, it is usually required that the node voltage, current, power, and permeability satisfy the constraints after DGs are connected to the grid.

a. Node voltage constraints

$$U_i^{\min} \le U_i \le U_i^{\max} \tag{12}$$

where $U_i$ represents the voltage amplitude of the $i$th node, and $U_i^{\min}$ and $U_i^{\max}$ represent the upper and lower limits of the $i$th nodes voltage amplitude, respectively.

b. Branch current constraints

$$I_{ij} \le I_{ij}^{\max} \tag{13}$$

where $I_{ij}$ is the current of branch $(i, j)$ and $I_{ij}^{\max}$ is the maximum current allowed of branch $(i, j)$.

c. Single DG access capacity constraints

$$\begin{cases} P_{DGi}^{\min} \le P_{DGi} \le P_{DGi}^{\max} \\ Q_{DGi}^{\min} \le Q_{DGi} \le Q_{DGi}^{\max} \end{cases} \tag{14}$$

where $P_{DGi}$ and $Q_{DGi}$ are the active power and reactive power of the DG connected to the $i$th node, respectively.

d. Distributed generation access total capacity constraints

$$\sum_i^N S_{DGi} \le 0.3 S_L \tag{15}$$

where $\sum S_{DGi}$ is the total capacity of distributed generation and $S_L$ is the maximum load value of DN.

e. Short-circuit current constraints

$$\begin{cases} K_{sen,i}^{\mathrm{III}} = \dfrac{I_{k,i+1.\min}^{(2)}}{I_{set,i}^{\mathrm{III}}} \ge 1.2 \\ I_{set,i}^{\mathrm{I}} > I_{k,i.\max}^{(3)} \\ I_{set,i}^{\mathrm{I}} > I_{rk,i-1.\max}^{(3)} \\ I_{set,i}^{\mathrm{II}} > I_{rk,i-1.\max}^{(3)} \\ I_{set,i}^{\mathrm{III}} > I_{rk,i-1.\max}^{(3)} \\ I_{set,j}^{\mathrm{I}} > I_{k,j.\max}^{(3)} \end{cases} \tag{16}$$

where $K_{sen,i}^{\mathrm{III}}$ is the sensitivity coefficient of current III segment protection of branch $i$ as backup protection; $I_{set,i}^{\mathrm{I}}$, $I_{set,i}^{\mathrm{II}}$, and $I_{set,i}^{\mathrm{III}}$ are the setting values of the current I, II, and III protections of branch $i$, respectively; $I_{set,j}^{\mathrm{I}}$ is the setting value of the current I segment protection of the branch $j$ of the feeder line of the DG; $I_{k,i-1.\max}^{(3)}$ is the maximum reverse current through branch $i$ when three-phase short circuit occurs at the end of the upstream of branch $i$; $I_{k,i.\max}^{(3)}$ and $I_{k,j.\max}^{(3)}$ are the maximum short-circuit currents through branch $i$ of the terminal three-phase short circuit of branch $i$ and branch $j$, respectively; and $I_{k,i+1.\max}^{(2)}$ is the minimum short-circuit current through branch $i$ of the two-phase short circuit at the end of the downstream of branch $i$.

3. Overview formulation

From the above analysis, the multi-objective function and constraints can be described as follows:

$$\min\left[f_1(x_s, x_c), f_2(x_s, x_c), \cdots, f_{N_{obj}}(x_s, x_c)\right] \tag{17}$$

$$s.t. \quad h(x, x) = 0, i = 1, \cdots, p \tag{18}$$

$$g_i(x_s, x_c) \leq 0, \ i = 1, 2, \cdots, q \tag{19}$$

where $f_i$ is the $i$th objective function, $N_{obj}$ is the number of objective functions, $x_s$ and $x_c$ denote the state vector and the control vector respectively, $h_i$ is the equation constraint, $p$ is the number of equality constraints, $g_i$ is the inequality constraint, and $q$ is the number of inequality constraints.

$x_c$ is composed of $n$ variables, and $n$ represents the number of nodes of the optimized network, where each variable represents two components: the DG installation placement and capacity. For example, a variable is assigned a value indicating the placement of the variable to installation DG. The value of the variable indicates the installation capacity. If DG is not installed, the corresponding variable value is 0. $x_c$ can be illustrated as follows:

$$x_c = [(L_{DG1}, P_{DG1}), (L_{DG2}, P_{DG2}), \cdots, (L_{DGN}, P_{DGN})] \tag{20}$$

where $L_{DGi}$ indicates that the $i$th node is the DG installation location and $P_{DGi}$ represents the DG installation capacity of the $i$th node.

$x_s$ is the network parameter that $x_c$ is calculated from the power flow. Variables such as line losses, node voltage, and branch current are used to calculate multi-objective functions and to determine the satisfaction of constraints. $x_s$ can be illustrated as follows:

$$x_s = [P', Q', U', I', \theta'] \tag{21}$$

where $P', Q', U', \theta'$, and $I'$ represent the network feeder active power vector, reactive power vector, node voltage vector, voltage argument vector, and branch current vector respectively after DG is connected.

### 2.4. Establish PQ Mode of DG

The DG injects its power output into the grid through power electronics. Typically, for a PQ model, which shows the active power (P) versus reactive power (Q) called a PQ model. DG is modeled as a constant power factor and negative load model.

In this case, the DG is modeled as a constant power source. $P_{DG}$ is the actual power output specified for the DG model. The load at $i$th node with the DG device is modified as Equations (22)–(23).

$$P'_{loadi} = P_{loadi} - P_{DGi} \tag{22}$$

$$Q_{DGi} = P_{DGi} \cdot \tan\left(\cos^{-1}(\varphi)\right) \tag{23}$$

where $P_{loadi}$ is the original network load power of $i$th node, $P_{DGi}$ and $Q_{DGi}$ are the active and reactive power of DG at the $i$th node, $P'_{loadi}$ is the active power after installing DG, and $\cos\varphi$ is the power factor.

In general, constrained problems can be solved using either deterministic or stochastic algorithms. However, deterministic approaches such as feasible direction and generalized gradient descent require strong mathematical properties of the objective function such as continuity and differentiability. In cases where these properties are absent, evolutionary computation, such as NSGA-II, offers reliable alternative methods [13].

## 3. Improved NSGA-II Algorithm

### 3.1. NSGA-II Algorithm Overview

NSGA-II was proposed by Deb, K. in 2002 [28], which is one of the most popular multi-objective optimization algorithms. It uses simulated binary crossover and polynomial variation to introduce non-dominated sorting and crowding distance operator instead of NSGA. The sharing radius of the algorithm ensures the diversity of the population and the uniformity of the Pareto-optimal set and introduces the elite retention strategy and the elimination strategy, which improves the operation speed and stability of the algorithm.

### 3.2. Dominant, Non-Dominated, Pareto-Optimal Set and Crowding Distance

NSGA-II introduces Pareto-optimal set and crowding distance to replace the fitness of traditional intelligent optimization algorithm. The solution set is divided into dominated solution set and non-dominated solution set according to the dominant relationship among the solutions. The rank of the solution is from 1 to n: the better the quality, the smaller the value. Solutions of the same rank are divided into pros and cons by comparing the size of the crowding distance.

Multi-objective optimization problems can be described as follow:

$$\min f_i(x), \quad i = 1, 2, \ldots, N_{obj}, \; x \in I \tag{24}$$

where I is a feasible solution space.

**Definition 1.** *If solution $x_1$ dominates $x_2$, then the following definitions must be met.*

$$\forall i, j \in \left[1, 2, \cdots, N_{obj}\right], \\ \underset{i \neq j}{\exists} f_i(x_1) < f_i(x_2) \vee f_j(x_1) = f_j(x_2) \tag{25}$$

**Definition 2.** *If solution $x_1$ does not dominate $x_2$, then the following definitions must be met.*

$$\forall i, j \in \left[1, 2, \cdots, N_{obj}\right], \\ \underset{i \neq j}{\exists} f_i(x_1) \leq f_i(x_2) \wedge f_j(x_1) > f_j(x_2) \tag{26}$$

Equations (25) and (26) in Definitions 1 and 2 are derived from Reference [29].

**Definition 3.** *Pareto-optimal set is composed of mutually non-dominated solution sets, and ranks are composed of mutually dominated solution sets.*

**Definition 4.** *Assume that $P = \{x_1, x_2, \cdots, x_n\}$ is a Pareto-optimal set and that the crowding distance represents the distribution density of other solution sets around the solution. The larger the solution distance at the same layer, the better the solution distribution, that is, the better the solution set diversity. Its calculation formula is as follows:*

$$cd_i^k = \begin{cases} \frac{f_{i+1}^k - f_{i-1}^k}{f_{max}^k - f_{min}^k}, if \; index\left(x_i^k\right) \in [2, n-1] \\ \infty, \qquad otherwise \end{cases} \tag{27}$$

$$cd_i = \sum_{k=1}^{m} cd_i^k \tag{28}$$

*where $cd_i^k$ is the crowding distance on the kth objective function of $x_i$, m is the number of objective functions, index $(x_i^k)$ is the sort index of $x_i$ on the kth objective function, $f_{i+1}^k$ is the value of objective function corresponding to the last solution on the axis of the kth objective function, and $f_{max}^k$ and $f_{min}^k$ represent the maximum and minimum values of the kth objective function, respectively.*

### 3.3. INSGA-II: Improved Mutation Operator

Because mutations in genetics often lead to worse results, the probability of mutation operations in genetic algorithm (GA) is usually set to be very small. In GA, crossover and mutation operations have two functions: global search and local search. When dealing with low-dimensional convex problems, only the crossover operator can solve these problems well. However, when dealing with

high-dimensional nonconvex problems, it often falls into the local optimal solution. For this reason, the mutation operator is improved by referring to the population-updating method in the fireworks algorithm [25]; the new mutation individual is created by Equations (29)–(31).

$$d_i = vd \cdot rand \tag{29}$$

$$S = randperm(1, d_i) \tag{30}$$

$$u'_j = \begin{cases} u_j^{mu}, if \ j \in S \\ u_j, otherwise \end{cases} \tag{31}$$

where $d_i$ denotes the dimension of the $i$th solution for mutation, $vd$ denotes the variable dimension, *rand* denotes the random number between [0, 1], $S$ denotes the index for performing the mutation operation, $u_j^{mu}$ denotes the $j$th gene performing the mutation operation, and $u'_j$ represents the value of the $j$th gene after the mutation operation.

Crossover and mutation operations often produce solutions beyond the feasible space. The general method is to assign solutions less than the lower limit of feasible region to the lower limit, and the solution that exceeds the upper limit of the feasible region is given the upper limit value. This operation is simple to perform but reduces the diversity of solution sets. According to Equation (32), solutions beyond the boundary can be effectively mapped back to the feasible space.

$$\hat{u}_i = u_i^{\min} + |u_i|\%\left(u_i^{\max} - u_i^{\min}\right) \tag{32}$$

where $u_i^{\max}$ and $u_i^{\min}$ represent the upper limit and lower limit of the gene, respectively, and % is the remainder operation.

### 3.4. INSGA-II: Violation Constrained Index

In view of the subjectivity of penalty factor selection, the violation constrained index (VCI) is proposed when dealing with infeasible solutions.

The method divides the solution set into feasible solution and infeasible solution by calculating the error degree (ED) of the constraint condition of the solution.

The VCI calculation formula is as follows:

$$ED(x_s, x_c) = \left|h(x_s, x_c)\right| + \max(g(x_s, x_c), 0) \tag{33}$$

$$VCI_i = \sum_j^C \frac{ED_j^i - ED_j^{\min}}{ED_j^{\max} - ED_j^{\min}} \tag{34}$$

where $ED_j^{\max}$ and $ED_j^{\min}$ represent the maximum and minimum error degree of infeasibility under the $j$th constraint, respectively, and $C$ indicates the number of constraints. The classification process of the solution set is performed according to Equation (35).

$$S\_t = \begin{cases} 0, \ if \ \text{all} \ VCI_i = 0 \\ 1, \ if \ \text{all} \ VCI_i \mathrel{!=} 0 \\ \frac{1}{2}, \ otherwise \end{cases} \tag{35}$$

where $S\_t$ denotes the type of the solution set. The default index can be effectively used in combination with the non-dominated sorting strategy. For example, when the $S\_t$ is 0 or 1/2, the feasible solution part is given the true rank and the crowding distance, and the rank of the infeasible solution is sorted in descending order according to $VCI_i$ and continues to be sorted from the feasible solution.

### 3.5. INSGA-II: Improved Crowding Distance Operator

Figure 2 shows that, after solutions a–h pass the truncation strategy, a, d, e, g and h are preserved but the eliminated solution c has a better crowding distance than the solution d, which is due to the traditional crowding distance information being missing, and the truncation strategy eliminates the solution set with small crowding distances at one time, resulting in uneven distribution of the final set. The calculation of the traditional crowding distance only considers the spatial distribution of two adjacent solutions on the axis of the objective function without considering that the effect of deleting the neighborhood solution leads to the elimination of potential high-quality solutions. Finally, a potential crowding distance is proposed. The potential crowding distance in the two-dimensional space is simple to calculate. As shown in Figure 2a, when the solution map is re-target functions 1 and 2, the neighborhoods are the same in the solution set and only the order changes. The truncation process is shown in Figure 2b,c:

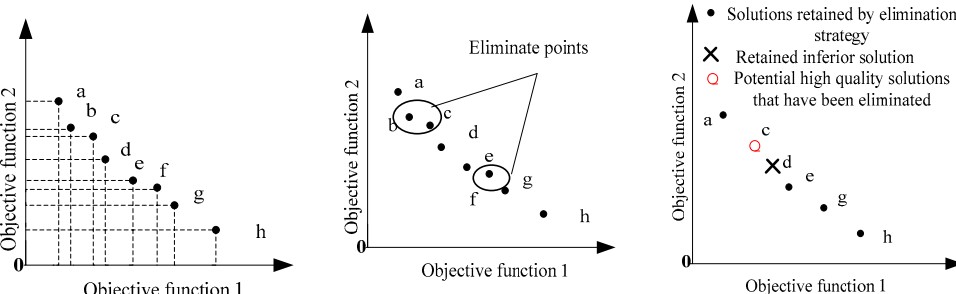

(**a**) Individual distribution chart　　(**b**) Eliminating points chart　　(**c**) Truncation process chart

**Figure 2.** Pareto-optimal set of truncation strategy.

Therefore, the formula for calculating the potential crowding distance in two-dimensional space is as follows:

$$
\begin{cases}
\Delta d_i^1 = \dfrac{f_{i-1}^1 - f_{i-2}^1}{f_{\max}^1 - f_{\min}^1} + \dfrac{f_{i+2}^2 - f_{i+1}^2}{f_{\max}^2 - f_{\min}^2}, \\[2mm]
\Delta d_i^2 = \dfrac{f_{i+2}^1 - f_{i+1}^1}{f_{\max}^1 - f_{\min}^1} + \dfrac{f_{i-1}^2 - f_{i-2}^2}{f_{\max}^2 - f_{\min}^2}, \\[2mm]
pd_i = cd_i + \max\left(\Delta d_i^1, \Delta d_i^2\right)
\end{cases}
\tag{36}
$$

where $\Delta d_i^1$ and $\Delta d_i^2$ denote the increments of the crowding distance after the adjacent solution of the *k*th solution is deleted, $f_{i-1}^1$ and $f_{i-1}^2$ denote the neighborhood solutions mapped to the *k*th solution on the objective function 1, and $pd_i$ denotes the potential crowding distance of the *k*th solution.

The objective function above three-dimensions does not have the symmetric relationship of the two-dimensional objective function. Each solution no longer has only the same two adjacent solution sets on each objective function axis but has (2–n) different solutions. Therefore, the potential crowding distance of the solution under the 3D objective function is calculated as follows:

$$
pd_i = cd_i + \max(k\_\Delta d_i), k \in \Phi
\tag{37}
$$

$$
k\Delta d_i = \sum_{j=1}^{m} 1^j \left[ 1^l \left( \frac{f_{i-1}^j - f_{i-2}^j}{f_{\max}^j - f_{\min}^j} \right) + 1^r \left( \frac{f_{i+2}^j - f_{i+1}^j}{f_{\max}^j - f_{\min}^j} \right) \right]
\tag{38}
$$

where $\Phi$ denotes the set of adjacent fields mapped to the *i*th solution on each objective function axis; *m* denotes the number of objective functions; $k\Delta d_i$ denotes the increment generated by the crowding distance of the *i*th solution after the *k*th solution in the neighborhood set is eliminated; and $1^j$, $1^l$, and $1^r$ respectively indicate that the *k*th solution belongs to the neighborhood of the *i*th solution or belongs to the left neighborhood solution or the right neighborhood solution on the *j*th objective function.

## 3.6. INSGA-II: Improved Crowding Distance

NSGA-II uses the tournament selection operator for population reproduction, and the selection criteria considers the rank and crowding distance. The specific selection process is described in Reference [27]. Since the improved NSGA-II algorithm increases the potential crowding distance, it needs to involve the crowding distance for the selection operation. After some modifications, the selection rules are as follows:

$$
newx_k = \begin{cases} x_i, if\ cd_i > cd_j\ and\ pd_i > pd_j \\ x_j, if\ cd_j > cd_i\ and\ pd_j > pd_i \\ x_i\ or\ x_j, otherwise \end{cases} \tag{39}
$$

where $newx_k$ denotes the selected $k$th descendant solution, and $x_i$ and $x_j$ respectively denote two different solutions in the parent. Improved selection strategy comprehensively considers the current crowding distance and potential crowding distance. Therefore, it can select a better solution and can avoid the defects of insufficient diversity of the solution set when the truncation strategy is executed.

## 3.7. Unbiased Compromise Strategy

In order to solve the cumbersome scheme of the Pareto-optimal set, this paper uses an unbiased compromise strategy based on fuzzy set, which uses unbiased member parameter $\omega$ to evaluate the quality of each individual in the Pareto-optimal set. The calculation formula is as follows:

$$
\omega^* = \max_{i=1,\cdots,n} \left( \frac{\sum_{j=1}^{m} \omega_j^i}{\sum_{i=1}^{n} \sum_{j=1}^{m} \omega_j^i} \right) \tag{40}
$$

where $\omega_j^i$ denotes the unbiased parameter of the $i$th solution in the Pareto-optimal set on the $j$th objective function; $f_j^{\min}$ and $f_j^{\max}$ denote the minimum and maximum values of the $j$th objective function corresponding to the Pareto-optimal set, respectively; $f_j^i$ denotes the value of the $j$th objective function corresponding to the $i$th solution in the Pareto-optimal set; $n$ denotes the number of objective functions; $m$ denotes the number of solutions in the Pareto-optimal set; and $\omega^*$ denotes the unbiased member parameters of the comprehensive optimal solution.

## 3.8. Optimized Configuration Process Based on Improved NSGA-II

The proposed algorithm optimizes the placement and sizing process configuration of DG, as follows:

---

**Algorithm 1. The algorithm of the proposed INSGA-II**

---

Input: Set IEEE-33 distribution network-related parameters, the number of target functions, the number of variables, the size of the population, the maximum number of iterations, etc.

Outputs: Comprehensive optimal solution $x^*$.

1: Initialization population $P(x_c)_t$, set the number of iterations t = 0;

2: $P(x_s)_t$ is calculated by forward and backward substitution method, and then, the objective function values $f_1$, $f_2$, $f_3$, and $S\_t$ are calculated;

3: Algorithm iteration start, while $t < t_{\max}$ do;

4: The solution set is classified by $S\_t$, feasible solution to perform non-dominated sorting strategy, and the infeasible solution to calculate $DCI$ and sorted;

5: Generation of children $Q(x_c)_t$ from the parent $P(x_c)_t$ by improving selection operations and improving genetic manipulation;

6: $Q(x_c)_t$ performs power flow calculation, mixes with $P(x_c)_t$ to form a mixed population $R(x_c)_t$ and $R(x_c)_t$ for non-dominated sorting strategy, and calculates the crowding distance according to Equations (37) and (38);

7: Selecting a new parent population $R(x_c)_t$ of size from $P(x_c)_{t+1}$ by a truncation strategy;

8: $t = t + 1$;

9: End while;

10: Using unbiased compromise strategy to select comprehensive optimal solution $x^*$;

11: Return $x^*$.

---

## 4. Experiment and Results

### 4.1. Experiment Setup and Description

In order to better verify the effectiveness of the proposed algorithm, IEEE 33-, 69-, and 118-bus systems are introduced as simulation examples to compare performance with current mainstream multi-objective optimization algorithms, considering the dynamic and static characteristics of the DGs and DN. Hardware parameters used in the experiment are as follows: Intel (R) Core (TM) i5-3337 CPU 1.80 GHz, memory: 4.00 GB, and simulation software: Matlab-2014a (The MathWorks, Inc.3 Apple Hill Drive, Natick, MA, USA, 2014).

### 4.2. IEEE 33-Bus System Case Simulation Experiment

An IEEE 33-bus system topology diagram is shown in Figure 3, where the system includes 33 nodes and 32 branches [30]. The 0 node is assumed to be a balanced node with a voltage reference of 12.66 kV, total power load of 3.17 MW, and reactive power 2.30 MVAr.

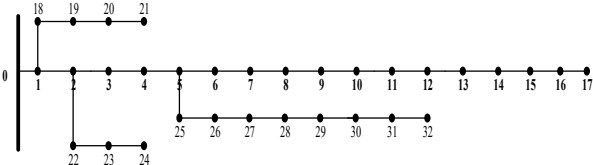

**Figure 3.** IEEE 33-bus system topology diagram.

Multi-objective function parameter setting: The maximum installation capacity of each node of DG is 1 MW, the annual maximum load utilization time and the annual maximum line loss hour of DGs are 4500 h, the investment cost of micro-turbine (MT) unit equipment is 0.07 million $/kW, annual operation and maintenance cost is 0.009 $/kWh, wind turbine generator (WTG) group unit equipment investment cost is 0.114 million $/kW, annual operation and maintenance costs are 0.004 $/kWh, MT and WTG unified as a PQ model for processing, power factor is 0.9, the unit price of the loss is 0.071 $/kWh, the financial subsidy per unit of electricity is 0.019 $/kWh, and the conversion factor of the annual equipment investment cost of the distributed power source is obtained by 3% of the annual interest rate. We use the algorithm of the proposed INSGA-II for the following series of analysis.

#### 4.2.1. Analysis of Optimization Results with Different Parameters

As shown in Table 1, in order to improve the operation speed of the algorithm and the accuracy of the solution, experimental comparisons of the experimental hyperparameters are made. Compared with experiment 1 and experiment 2, the optimization results are improved with the increase of iteration times.

**Table 1.** Simulation results of INSGA-II with different parameters.

| Experiment | Population Size (NP) | Iterations (tmax) | Time (s) | Energy-Saving Benefit (Million $) | Voltage Deviation (p.u) | Line Loss (MW) |
|---|---|---|---|---|---|---|
| 1 | 50 | 100 | 26.71 | 0.193 | 5.289 | 0.081 |
| 2 | 50 | 200 | 52.46 | 0.196 | 5.273 | 0.084 |
| 3 | 100 | 100 | 83.67 | 0.197 | 5.254 | 0.082 |
| 4 | 200 | 100 | 107.91 | 0.199 | 5.267 | 0.081 |

Comparing experiments 1, 3, and 4, the optimization effects are improved with the increase of population size but the time increases. The population selection of 100 and the number of iterations of 100 are taken as the hyperparameters of the algorithm.

Through large numbers of experiments, subsequent experiments decided to set improved NSGA-II parameters: population size is 100, number of iterations is 100, and crossover probability is 0.7.

4.2.2. Analysis of DG Installation Capacity Optimization Results

The DG optimization configuration results of the improved NSGA-II are shown in Table 2.

**Table 2.** INSGA-II optimization results.

| Case | DG Location (Bus Number) | DG Sizes (MW) | Energy-Saving Benefit (Million $) | Voltage Deviation (p.u) | Line Loss (MW) |
|------|--------------------------|---------------|-----------------------------------|-------------------------|----------------|
| 1 | - | - | 0 | 19.6024 | 0.2017 |
| 2 | 5/6/17/32 | 0.2379/0.2391/0.6965/0.4451 | 0.1976 | 5.2453 | 0.0797 |
| 3 | 6/17/24/32 | 0.2148/0.5013/0.2062/0.4126 | 0.1781 | 4.9348 | 0.0613 |

As shown in Table 2, Case-1 has no DG, Case-2 is installation without short-circuit current constraints (other constraints are considered), and Case-3 is installation with short-circuit current constraints (other constraints are considered). Compared with Case-1, the planning of Case-2 and -3 obtained by INSGA-II can effectively improve the objective function. Case-2 can produce energy-saving benefits of about $0.197 million, the voltage deviation is improved by 73.24%, and the line loss is reduced by 69.60%. Case-3 can produce energy-saving benefits of about $0.193 million, the voltage deviation is improved by 74.83%, and the line loss is reduced by 60.49%. Comparison of Case-2 and -3 show that, under the influence of short-circuit current protection constraints, the capacity of DGs decrease and voltage deviation and line losses are improved. In addition, when the DGs are close, the short-circuit current of some branches will be very high, so the Case-3 optimization results do not show that the DG access points are close. The above results confirm the rationality of considering short-circuit current constraints.

As shown in Figure 4a,b, Case-2 and -3 can improve the voltage amplitude of each node and line losses by installing DG appropriately. Because most of the installation nodes of Case-3 are terminal nodes of the system, the supporting effect of the node voltage is stronger [31]. Active network loss is usually determined by node voltage and branch resistance. Distribution network voltage level is lower and R/X value is larger, so it will lead to larger network losses. When DG is reasonably connected, the voltage fading will be improved [31]. In the case of the conditions, Case-3 has a greater effect on node voltage support than Case-2, so the line losses are better.

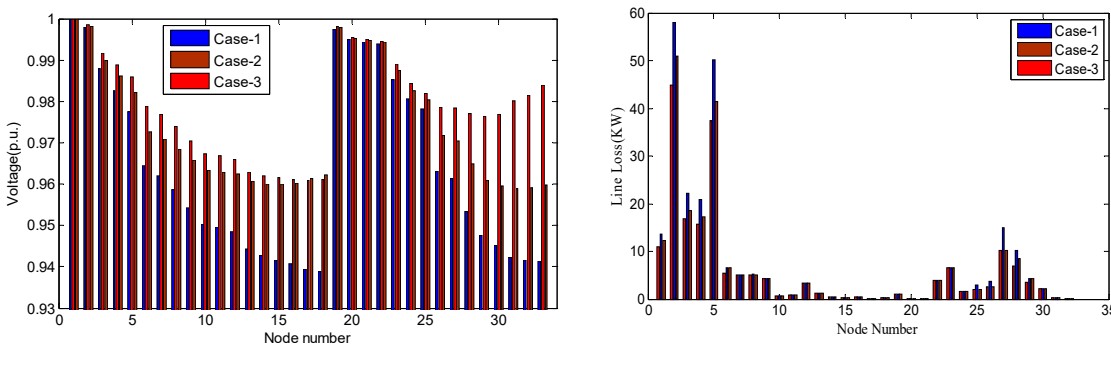

(**a**) The voltage amplitude of each node      (**b**) The line loss of each node

**Figure 4.** Results of nodal voltage optimization.

*4.3. Performance Analysis of INSGA-II Algorithm*

In order to verify the optimization performance of the proposed algorithm, INSGA-II, NSGA-II, and MOPSO are used to analyze and compare the planning results of the IEEE 33-bus system

independently. The parameters of NSGA-II are set as INSGA-II, the inertia factor is 0.8, the local speed factor is 0.1, and the global speed factor is 0.1; other parameters are the same as INSGA-II.

As shown in Figure 5, NSGA-II and MOPSO have uneven distribution of Pareto solutions due to the defects of crowding distance operators. MPSO optimizes by choosing leaders and by archiving mechanism. Similar to NSGA-II, its selection process also depends on a dominant relationship, so crowding distance is also needed to participate in it. When solving the high-dimensional nonconvex problem, the solution set is easy to fall into the local optimal solution and the population diversity is insufficient. Obviously, the uniformity of Pareto-optimal set obtained by INSGA-II is the best of the three algorithms, which verifies the validity of the potential crowding distance and improves the mutation operator to improve the optimization ability of the algorithm.

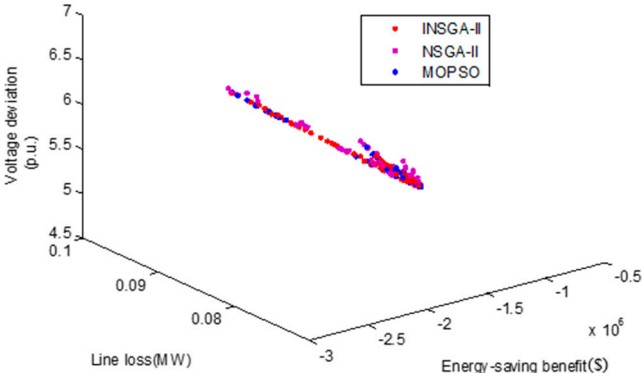

**Figure 5.** Pareto-optimal set of three algorithms.

In order to verify the performance of the improved algorithm more intuitively, the convergence metric [29] and the spacing metric [32] are used to compare the three optimization algorithms.

(1) C_mertic

Convergence metric can be expressed as follows:

$$I_C(X', X'') = \frac{|\{a'' \in X''; \exists a' \in X' : a' \prec a''\}|}{|X''|} \tag{41}$$

where $X'$ and $X''$ are respectively two solution sets, $a'$ and $a''$ are solutions belonging to two solution sets, $p$ is the dominant symbol, and $|X''|$ is the number of solutions in the solution set $X''$. If $I_C(X', X'')$ is 1, all solutions in the solution set $X'$ dominate the solution in the solution set $X''$, and if $I_C(X', X'')$ is equal to 0, all solutions in the solution set $X''$ are not dominated by the solution set $X'$.

As shown in Table 3, about 32.39% of NSGA-II and 33.04% of MOPSO are dominated by INSGA-II. In contrast, only about 6.41% and 7.92% of INSGA-II are dominated by NSGA-II and MOPSO, respectively. In addition, the computational complexity O ($mn^3$) is similar, so the running time of the three algorithms is almost the same, which proves that the improved crowding distance operator can effectively improve the quality of the solution set and can ensure operation efficiency.

**Table 3.** Convergence metric of each algorithm.

| Algorithm | NSGA-II | INSGA-II | MOPSO | Time(s) |
|---|---|---|---|---|
| NSGA-II | - | 6.41% | 11.21% | 52.87 |
| INSGA-II | 32.39% | - | 33.04% | 53.04 |
| MOPSO | 15.73% | 7.92% | - | 53.12 |

(2) S_mertic

Spacing metric [27] can be expressed as follows:

$$I_S = \sqrt{\frac{1}{N-1}\sum_{i=1}^{N}\left(\bar{d}-d_i\right)^2} \tag{42}$$

where $N$ represents the number of solutions, $d_i$ represents the shortest distance from the $i$th individual to the rest of the solution, and $\bar{d}$ represents the mean of all individuals $d_i$. A smaller spacing metric means that the solution distribution in the Pareto solution is more uniform. The zero value of the interval metric means that all solutions in the Pareto-optimal set are equally spaced.

As shown in Figure 6, in order to visually describe the uniformity of the distribution of the solution sets of different algorithms, the diversity indices obtained by each algorithm running 30 times independently are represented by box diagrams, in which each box represents the distribution of the measurement values of the distance between the solution sets of different algorithms. The upper quartile line at the top of each box diagram and the lower quartile line at the bottom represent the boundary values except outliers. If there is an abnormal value, use "+" to identify it. The median value is the red line in the rectangle box. Compared with the other two algorithms, INSGA-II has smaller median values (see red line in the middle) and minimum values (see quartile line below). Therefore, it is further explained that the distribution of INSGA-II solutions is more uniform and has better diversity.

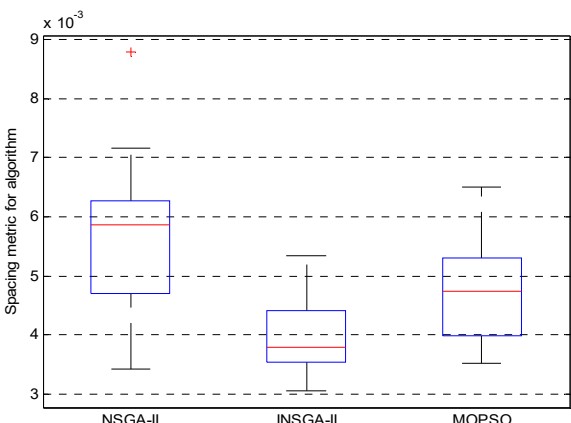

**Figure 6.** Three algorithms' spacing metric.

(3)　Analysis of the relationship between objective functions

As shown in Figure 7a, the energy-saving benefit has a nonlinear relationship with line loss, and the degree of line loss improvement will have an extreme point, which will not decrease with the increase of DG capacity but will be damaged. In view of Figure 7b, the voltage increases with the increase of energy-saving benefits. The results of DG optimization show that the capacity of DG-mounted nodes is too large to reduce the total voltage deviation correction effect. Therefore, the energy-saving benefits have a linear relationship with the voltage deviation when the permeability of DG is satisfied. Line loss and voltage deviation also show a nonlinear relationship in Figure 7c, and the two restrict each other.

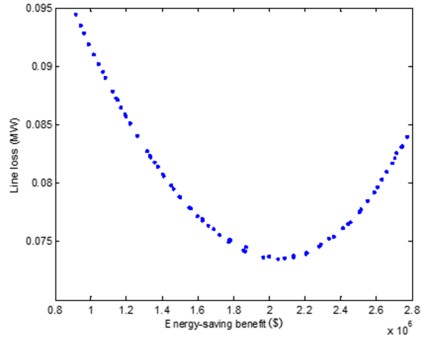

(**a**) The energy-saving benefit relationship with line loss

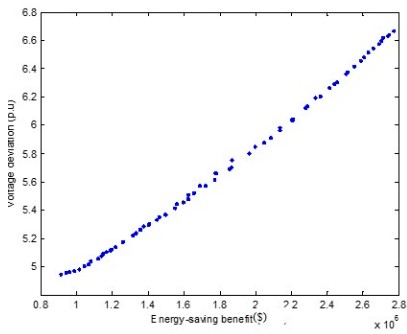

(**b**) The energy-saving benefit relationship with voltage deviation

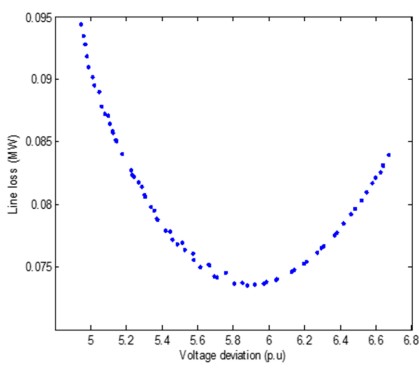

(**c**) The line loss relationship with voltage deviation

**Figure 7.** Relationship between different objective functions.

### 4.4. Case Analyses of IEEE 33-, 69-, and 118-Bus Systems

In order to prove the improvement degree of the NSGA-II algorithm on network optimization results of different node systems, the IEEE 33-, 69- and 118-bus systems are introduced. The parameters of the IEEE 69- and 118-bus systems are described in References [33,34].

Table 4 shows the optimization results of the INSGA-II algorithm on three examples. For the objective function, with the increase of network complexity, the improvement degree of the objective function of the three DNs increases. For example, IEEE 33-bus, 69-bus, and 118-bus are improved by 74.38%, 77.39%, and 77.32%, respectively, in terms of voltage deviation. On the hand, it also proves the importance of reasonable installation of DG. On the other hand, most of the installation nodes are the end of the system, which also proves that DGs can improve the voltage deviation better than other nodes.

**Table 4.** Results of different network optimization.

| Test System | DG Location (Bus Number) | DG Sizes (MW) | Optimal Objectives | | | |
|---|---|---|---|---|---|---|
| | | | Optimization | Energy-Saving Benefit (Million $) | Voltage Deviation (p.u) | Line Loss (MW) |
| 33-bus | 6/17/24/32 | 0.2148/0.5013/0.2062/0.4126 | Before/ | 0 | 19.6024 | 0.2017 |
| | | | After | 0.1781 | 4.9348 | 0.0797 |
| 69-bus | 27/35/39/56/52 | 0.4266/0.2618/0.3446/0.5426/0.5698 | Before/ | 0 | 31.6747 | 0.2206 |
| | | | After | 0.4876 | 7.1614 | 0.0609 |
| 118-bus | 3/17/27/18/80/116 | 0.7787/0.4544/0.7240/0.6483/0.1027/0.2698 | Before/ | 0 | 54.3018 | 0.6427 |
| | | | After | 0.8161 | 12.3112 | 0.10272 |

### 4.5. Experiments on Load and DG Output Considering Annual Time Series Changes

With the increase of installations and capacity of DG in the future, considering only static load and DG operation status, it may not be applicable to other load levels of permeability constraints and may even lead to reverse power flow, voltage limit, original system protection measure failures, and other phenomena, so that the system is threatened.

To this end, this section optimizes the configuration scheme to consider the DG annual output change and load change trend and the revised IEEE 33-bus system-related data reference [34]. The data was recorded every hour, so it is assumed that the DG output is equal every hour and that the load demand is constant. The four-season output level and load demand of wind turbines are shown in Figure 8a,b.

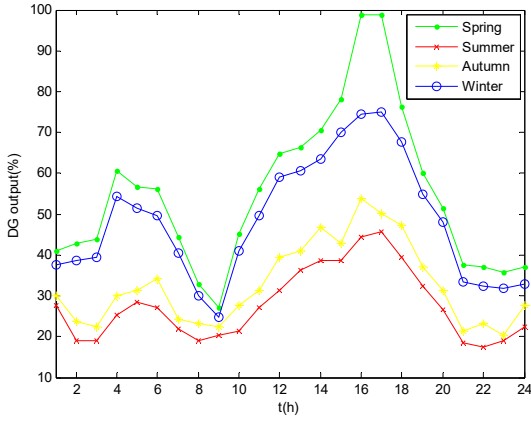

(**a**) The four-season output level

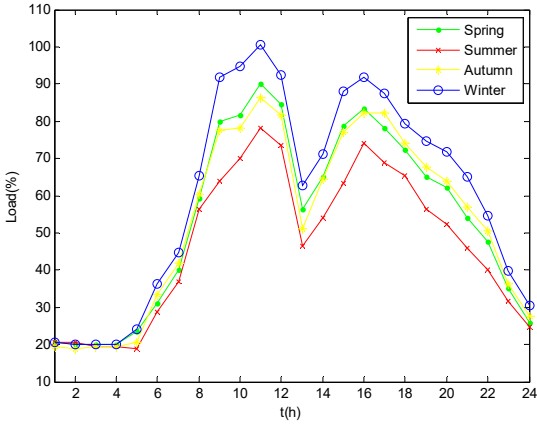

(**b**) The four-season load demand

**Figure 8.** Time series data of wind turbine output and load.

As seen in Figure 8a,b, the wind speed is higher in spring and winter and the output of wind turbine is also improved. On the contrary, the output of wind turbine in summer and autumn is small and the seasonal data of the load has changed but the overall change is not large. On the one hand, if the optimization is based on the summer and autumn season time data, it may lead to waste of DG capacity installation. On the other hand, optimizing the configuration in spring and winter will destroy the network penetration rate and cause economic and security crisis. Here, we choose to optimize the four seasons separately and consider them comprehensively. The four seasonal DG optimization results using INSGA-II are shown in Table 5.

**Table 5.** Four quarterly independent optimization results.

| Seasonal Condition | DG Location (Bus) | DG Sizes (MW) | Optimal Objectives | | | |
|---|---|---|---|---|---|---|
| | | | Optimization | Energy-Saving Benefit (Million $) | Voltage Deviation (p.u) | Line Loss (MW) |
| Spring | - | - | Before/ | 0 | 471.6464 | 1.5656 |
| | 6/17/24/32 | 0.2148/0.5013/0.2062/0.4126 | After | 0.0056 | 121.3572 | 1.5326 |
| Summer | - | - | Berfore/ | 0 | 472.3140 | 0.3733 |
| | 7/17/24/32 | 0.1962/0.6129/0.1135/0.4544 | After | 0.0065 | 67.9313 | 0.3599 |
| Autumn | - | - | Before/ | 0 | 472.2005 | 0.5149 |
| | 6/17/24/32 | 0.1308/0.5136/0.1012/0.4227 | After | 0.0067 | 83.6967 | 0.5019 |
| Winter | - | - | Before/ | 0 | 471.8158 | 1.1628 |
| | 6/17/24/32 | 0.2029/0.5364/0.1910/0.4313 | After | 0.0583 | 119.9812 | 1.1352 |

As shown in Table 5, due to the relatively stable output of the wind turbines in summer and autumn, it is possible to seek a relatively high-quality configuration result. The spring and winter seasons are relatively unsatisfactory because of the large fluctuations in output. The combined installation capacity of the dynamic optimization scheme should meet the DG permeability constraint of the remaining time period. In each season, it is quite difficult for a DG configuration scheme to improve the optimization function while satisfying the different constraints of 24 h. Therefore, the installation capacity of each season is almost close to the maximum installation capacity, which is a compromise that is comprehensively considered to optimize the time-series data of each objective function. In order to meet the annual operating constraints, the combination of the smallest installation capacity on each quarter node is selected as the planning scheme.

## 5. Conclusions

Considering the influence of DG access on the economy, security, and reliability of DN, this paper establishes a multi-objective optimization model to minimize the line losses and voltage deviation and to maximize the annual energy benefit. For DN with integrated DG units, equality constraints and inequality constraints, which include power-flow equality, nodal voltage, branch current, DG capacity, and short-circuit current, are considered in the optimization model. The principle of installation of DG is summarized through experiments and power system theory, and the violation constraint indicators are proposed to solve the infeasible solution.

The mutation operator, crowding distance operator, and selection operator of traditional NSGA II are improved to increase the population diversity and consistency in the optimal allocation. The introduction of an unbiased, compromised strategy can quickly give decision makers a comprehensive plan to weigh each goal.

The different examples of the IEEE 33-, 69-, and 118-bus systems are analyzed and optimized, and different super parameters are compared experimentally. The performance of the proposed INSGA-II is verified by comparing NSGA-II and MOPSO algorithms. The IEEE 69- and 118-bus systems are selected as the verification case. The proposed algorithm is applied to solve the static and dynamic characteristics of the DN with the integrated DGs. Finally, the IEEE 33-bus system is modified and

optimized by using the four season data of wind turbine and load. The results indicate that the proposed method can achieve better precision and diversity and can provide a good configuration plan for decision makers under the premise of meeting the annual penetration rate.

In practice, the choice of the best site may not always be feasible due to many reality constraints. However, the optimization and analysis here suggest that considering multi-objectives helps to decide siting and sizing of DG units for the decision maker. The optimization planning of a multi-distributed generation access distribution system combined with geographic information system (GIS) technology will be further investigated.

**Author Contributions:** Conceptualization, W.L. and F.L.; methodology, W.L.; software, F.L.; validation, Y.L., F.L., and W.D.; formal analysis, W.L.; investigation, F.L.; resources, F.L.; data curation, W.D.; writing—original draft preparation, F.L.; writing—review and editing, W.L. and Y.L.; visualization, W.D.; supervision, W.L.; project administration, W.L.; funding acquisition, W.L.

**Funding:** This research was funded by Natural Science Foundation of Heilongjiang Province, China, grant number E201332.

**Acknowledgments:** The authors would like to thank the editors and the reviewers for their constructive comments and suggestions.

**Conflicts of Interest:** The authors declare no conflict of interest. The funders had no role in the design of the study; in the collection, analyses, or interpretation of data; in the writing of the manuscript; or in the decision to publish the results.

## Abbreviations

| | |
|---|---|
| $Z_{cost}^{NODG}$ | total annual cost of DN without DGs |
| $Z_{cost}^{NODG}$ | total annual cost of DN with DGs |
| $C_{loss}$ | annual loss of DN without DGs |
| $C_b$ | annual purchase cost of DN without DGs. |
| $C_p$ | unit price of electricity consumed per unit |
| $\tau_{max}$ | annual maximum load loss hour (h) of the DN |
| $\Delta P_j$ | active power loss (kW) of the $j$th branch |
| $P_{load}$ | DN total active load |
| $P'_{loadi}$ | the active power after installing DG |
| $T_{max}$ | DN annual maximum load utilization hours |
| $C_{dgm}$ | annual investment and maintenance cost of DGs |
| $C_{ploss}$ | annual cost of DN with DGs |
| $C_B$ | annual purchase cost of DN with DGs |
| $C_{sub}$ | financial subsidy of new energy generation |
| $C_{dgi}$ | distributed generation equipment investment cost ($/kWh) |
| $C_{mi}$ | annual operation and maintenance cost ($/kWh) for the $i$th distributed generation |
| $P_{dgi}$ | installed capacity (kW) of the $i$th distributed generation |
| $C_{sp}$ | subsidy amount ($) of distributed generation unit |
| $g_{ij}$ | admittance of the branch $(i, j)$ |
| $U_i^{specified}$ | voltage-rated value |
| $P_{DGi}Q_{DGi}$ | $i$th node active and reactive power of the DGs injected |
| $P_{Li}Q_{Li}$ | $i$th node active and reactive powers of load output |
| $B_{ik}$ | the branch susceptance |
| $\sum S_{DGi}$ | the total capacity of distributed generation |
| $S_L$ | maximum load value of DN |
| $K_{sen,i}^{III}$ | the sensitivity coefficient of current III segment protection of branch $i$ as backup protection |
| $I_{set,i}^{I}$ | the setting values of the current I protection of the branch $i$, respectively |
| $I_{k,i-1.max}^{(3)}$ | the maximum reverse current through branch $i$ |
| $L_{DGi}$ | the $i$th node is the DG installation location |
| $cd_i^k$ | the crowding distance on the $k$th objective function of $x_i$ |
| $f_{i+1}^k$ | the value of objective function corresponding to the last solution on the axis of the $k$th objective function |

| $ED_j^{\max}$ | the maximum error degree of infeasibility under the $j$th constraint |
|---|---|
| $S\_t$ | the type of the solution set |
| $newx_k$ | the selected $k$th descendant solution |
| $\omega_j^i$ | the unbiased parameter of the $i$th solution in the Pareto-optimal set on the $j$th objective function |

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
