# Peer review of "Optimal Siting and Sizing of Distributed Generation Based on Improved Nondominated Sorting Genetic Algorithm II"

_processes, doi:10.3390/pr7120955_

Round 1
Reviewer 1 Report
Literature review in the area of problem domain must be improved -provide a more critical review with recent references to justify the research.
Main contributions are listed towards the end of section 1 - very good
The problem statement section should be improved by providing relevant examples to complement the equation explanations. Similarly, section 3.3 - 3.6 should be explained with examples. This is very important from a practitioner's viewpoint to understand the decision made while choosing various parameters.
e.g., tournament selection - why?; decisions on population size, number of experiments, etc.
Provide a list of most commonly used symbols to improve the readability
experiment results are easy to follow and are explained well
Improve the conclusion section by adding future work and tie the outcome based on the experiment results
the paper must be reviewed for spelling and grammar
Author Response
Dear Editor and Reviewer,
We are truly grateful to your critical comments and thoughtful suggestions. Based on these comments and suggestions, we have made careful modifications on the original manuscript.
1.Respond: Thanks you very much for the careful reading.
In Sections.3.3~3.6, the formulas 29~32 are derived from the fireworks algorithm, and the formulas 33~39 are the formulas derived by themselves according to the principle.
2.Respond:Most of the parameters related to intelligent algorithms are derived from experience and are parameters obtained after many experiments, so it is difficult to provide exact calculation formulas.The usual selection methods are the tournament selection method, the bet roulette method, etc., but since the bet round method relies on random numbers, the computer generates pseudo-random numbers, so it is not as good as the tournament selection method to some extent.
3.Respond:We will provide a list of the most commonly used symbols for readability.
4.Respond:We have improved the conclusions and results by adding future work and tie the outcome based on the experiment results.
5. Conclusions
Considering the influence of DG access on the economy, security and reliability of DN, this paper establishes a multi-objective optimization model to minimize the line losses and voltage deviation and maximize the annual energy benefit. For DN with the integrated DG units, equality constraints and inequality constraints,which include power-flow equality, nodal voltage, branch current, DG capacity , and short-circuit current are considered in the optimization model. The principle of installation of DG is summarized through experiments and power system theory, and the violation constraint indicators are proposed to solve the infeasible solution.
The mutation operator, crowding distance operator and selection operator of traditional NSGA II are improved to increase the population diversity and consistency in the optimal allocation. The introduction of unbiased compromise strategy can quickly give decision makers a comprehensive plan to weigh each goal.
The different examples of IEEE 33-, 69-, 118 systems are analyzed and optimized, and different super parameters are compared experimentally. The performance of the proposed INSGA-II is verified by comparing NSGA-II and MOPSO algorithms. IEEE 69-, 118-bus system is selected as the verification case. The proposed algorithm is applied to solve the static and dynamic characteristics of the DN with the integrated DGs. Finally, the IEEE 33-bus system is modified and optimized by using the four season data of wind turbine and load. The results indicate that the proposed method can achieve better precision and diversity , and provide a good configuration plan for decision makers under the premise of meeting the annual penetration rate.
In practice, the choice of the best site may not always be feasible due to many reality constraints. But the optimization and analysis here suggest that considering multi-objectives helps to decide placement and sizing of DG units for the decision-maker.The optimization planning of multi-distributed generation access distribution system combined with geographic information system (GIS) technology will be further investigated.
6.Respond:Thank you very much for your careful review of the manuscript format. We have modified the spelling and grammar of this paper exactly as required.

Reviewer 2 Report
Response of reviewer to the author.
The author has done some commendable work regarding DER aggregation model. The author has addressed a limitation in the existing literature and conducts a comprehensive scientific assessment. The authors have presented some good work and some comments are given below. I hope that if the authors will take them into account, the quality of the paper will be improved.
However, there are various points and queries, which need to be answered and addressed, respectively.
Q1. Lines 25-26 on page 1, the sentence has a typo error and is highlighted in red.
Most importantly, it also can. achieve desirable economic efficiency and safer voltage level.
Q2. Lines 34-38 on page 1, Kindly write distributed generation with (DG). The typo error is highlighted in red.
"research at home and abroad.After the DGs are connected to the distribution network (DN), the 34 structure, operation mode and control strategy of the DN will undergo tremendous changes [1-2]. 35 Research shows that DGS provides greater flexibility and scalability by connecting DN. By 36 optimizing the planning and operation of distributed generation, the power quality, reliability and 37 security of DN can be improved."
Q3. There are a plentiful of typo mistakes throughout the paper. Kindly correct them accordingly.
Q4. Kindly use few more credible references in the literature review, if you cannot add due to any reason then kindly give a reason. Further in-depth literature review is required. What are the limitations in objectives have addressed in this paper? Kindly improve the quality of literature review.
Q5. The authors does not mention that why they have considered INSGA-II algorithm only? What are the limitations that have expected to overcome left in the other approaches and have bridged with aforementioned proposed algorithm only?
Q6. Lines 135-143, page 3, why only these three objectives and whynot others, give a solid reason?
Q7. Lines 149-151, page 4, why RMB and why not USD? Does scaling factor have any impact on the change of currency? Is it region-centric or aims globally? Give reason.
Q8. Line 310-314, page 8, the typo error is highlighted in red.
The solution set is divided according to the dominant relationship between the solution and the solution.
Q9. Line 319-321, page 8, What does equation 25 and 26 in definition 1 and 2, refers to? Also how it can make it easy to the new reader, reading it?
Q10. 338-341, page 8, do you want to mention that you have improved mutation operator referring to population update method of firework algorithm? Kindly properly mention it.
Q11. Lines 330-333, Page 8,
Lines 345-348, Page 9,
Lines 364-366, Page 9,
Lines 368-371, Page 9,
Lines 389-407, Page 10,
Lines 409-412, Page 10,
Lines 419-423, Page 11,
Lines 442-449, Page 12,
Lines 511-514, Page 14,
Lines 517-528, Page 15,
The text shown in aforementioned lines throughout have different font. Kindly synchronize it with the main text. That is Palatino Linotype. Change it throughout the manuscript.
Q12. Lines 361-363, Page 9, have you derived equations 33 and 34 or have taken from any reference? Kindly mention it properly.
Q13. Lines 426, Page 11, what is DC1 and where it has shown before in the text?
Q14. Lines 442-449, Page 12, from where have you considered or taken this data? Kindly share or mention the reference.
Q15. Lines 450,463, Page 12,13, Give proper sub heading and do give space before start of new table, especially after second heading.
Q16. Lines 523-524, Page 15, have you derived equation 43 or have taken from any reference? Kindly mention it properly.
Q17. Lines 582-586, from where this seasonal data is taken? Give a proper reference.
Q18. How your work is different than the following works? Give solid and proper reason. There are also several other references as well.
Wanxing Sheng; Ke-Yan Liu ; Yuan Liu ; Xiaoli Meng ; Yunhua Li "Optimal Placement and Sizing of Distributed Generation via an Improved Nondominated Sorting Genetic Algorithm II", IEEE TRANSACTIONS ON POWER DELIVERY, VOL. 30, NO. 2, APRIL 2015, 569-578. Buayai, K.; Ongsaku,W.; Mithulananthan, N. Multi-objective micro-grid planning by NSGA-II in primary distribution system. Trans. Electr. Power 2011, 22, 170–187.

Author Response
Dear Editor and Reviewer,
We are truly grateful to your critical comments and thoughtful suggestions. Based on these
comments and suggestions, we have made careful modifications on the original manuscript. All changes made to the text are in red color. Below you will find our responses to the reviewer’s comments/ questions.
kind regards
Wei Liu

Reviewer 3 Report
This paper proposes an Improved non-dominated sorting genetic algorithm (INSGA-II) for solving the optimal siting and sizing of DG units. Firstly, the multi-objective optimization model is established by considering the energy -saving benefit, line loss and voltage deviation values. In addition, relay protection constraints are introduced on the basis of node voltage, branch current and capacity constraints. Secondly, the violation constrained index and improved mutation operator are proposed to increase the population diversity of NSGA-II, and the uniformity of the solution set of the potential crowding 18 distance improvement algorithm is introduced.
The work is well written, and well formulated. The results are good and are very interesting for the readers of the magazine.
This reviewer considers that the article can be published in the current form.
Author Response
We are truly grateful to your critical comments and thoughtful suggestions.
Best regards!
Wei Liu
Round 2
Reviewer 1 Report
Thank you for considering the reviews and making the suggested changes.
I still recommend considering comment #2 to improve the interest of the reader. "The problem statement section should be improved by providing relevant examples to complement the equation explanations. Similarly, section 3.3 - 3.6 should be explained with examples. This is very important from a practitioner's viewpoint to understand the decision made while choosing various parameters."
Also, going back to your response "Most of the parameters related to intelligent algorithms are derived from experience" - from an academic viewpoint - provided justification is not enough.
Author Response
Dear Reviewer,
We are truly grateful to your critical comments and thoughtful suggestions. Based on these comments and suggestions, we have made careful modifications on the original manuscript.
I still recommend considering comment #2 to improve the interest of the reader. "The problem statement section should be improved by providing relevant examples to complement the equation explanations.
Respond:A number of review papers have surveyed the optimization techniques for optimal DG planning in power distribution networks. The aforementioned review papers mainly focused on the discussion of various computational methods and metaheuristic algorithms.
The optimal planning of distributed generation sizing and siting is critical to ensure the operational performance of distribution network in terms of power quality, voltage stability, reliability and profitability.
In line 45:
A number of review papers[8-10] have surveyed the optimization techniques for optimal DG planning in power distribution networks. The aforementioned review papers mainly focused on the discussion of various computational methods and metaheuristic algorithms.
In line 183:
Lowering the line losses can improve the energy utilization efficiency of the energy-using equipment or process and it is also one of the important measures of energy saving. The objective function can be expressed as[31]:
In line 311:
In general,constrained problems can be solved using either deterministic, or stochastic algorithms. However,deterministic approaches such as feasible direction and generalized gradient descent, require strongmathematical properties of the objective function such as continuity and differentiability.In cases where these properties are absent, evolutionary computation, such as NSGA-II, offers reliable alternative methods[10]..
Similarly, section 3.3 - 3.6 should be explained with examples. This is very important from a practitioner's viewpoint to understand the decision made while choosing various parameters."
Respond: Thank you for your careful review of this article. 3.5 INSGA-II: Improved crowding distance operator.The truncation process is shown in Fig. 2 (a),(b) and (c).As this article is too long, so we explain the relevant examples and results in part 4.
Also, going back to your response "Most of the parameters related to intelligent algorithms are derived from experience" - from an academic viewpoint - provided justification is not enough.
Respond: I'm sorry we didn't make ourselves clear. If we want to get empirical guidance, that is, the population is large, and the number of iterations can be relatively small. If the number of iterations is large, the population can be set to be small, and the cross parameters can be set to be large in the early stage, and exponentially attenuated in the later stage.

Reviewer 2 Report
Response of reviewer to the author.
The reviewer is grateful for authors' response. However, there are various points and queries, which need to be answered and addressed, respectively.
Q1. Kindly use few more credible references in the literature review, if you cannot add due to any reason then kindly give a reason. Further in-depth literature review is required. What are the limitations in objectives have addressed in this paper? Kindly improve the quality of literature review. They are not highlighted properly.
Q2. Line 729, Pages 21-22. List of symbols or abbreviations are usually at the start or at the end before references section.
Q3. The authors do not mention why they have considered INSGA-II algorithm only? What are the limitations that have expected to overcome left in the other approaches and have bridged with the aforementioned proposed algorithm only?
Q4. How your work is different than the following works? Give solid and proper reason. There are also several other references as well.
Wanxing Sheng ; Ke-Yan Liu ; Yuan Liu ; Xiaoli Meng ; Yunhua Li "Optimal Placement and Sizing of Distributed Generation via an Improved Nondominated Sorting Genetic Algorithm II", IEEE TRANSACTIONS ON POWER DELIVERY, VOL. 30, NO. 2, APRIL 2015, 569-578. Buayai, K.; Ongsaku,W.; Mithulananthan, N. Multi-objective micro-grid planning by NSGA-II in primary distribution system. Trans. Electr. Power 2011, 22, 170–187.Note: Provide a composite response on a separate file.
Author Response
Dear Reviewer,
We are truly grateful to your critical comments and thoughtful suggestions. Based on these comments and suggestions, we have made careful modifications on the original manuscript. All changes made to the text are in red color.
For detailed reply, please refer to the attached document.
Kind regards
Wei Liu

Round 3
Reviewer 2 Report
The reviewer is grateful for the authors' response.
Author Response
Dear reviewer,
Thank you for the valuable and insightful comment. As per your valuable comments we have improved our present version of the article.
kind regards
Wei Liu